

# Role of hypothalamus function in metabolic diseases and its potential mechanisms

Xinyu Zhang[1,2], Jie Yang[1,2], Yilin Li[1,2], Yulin Li[1,2] and Guoqi Li[1,2]

[1] Beijing Anzhen Hospital of Capital Medical University, Beijing, China
[2] Beijing Institute of Heart Lung and Blood Vessel Diseases, Beijing, China

## ABSTRACT

The hypothalamus, a crucial neuroendocrine regulatory center, plays a significant role in the occurrence and development of metabolic diseases. Recent advances in molecular biology and imaging technology have facilitated a better understanding of the central role of the hypothalamus in the dysregulation of regulatory mechanisms. This review examines the involvement of hypothalamic nuclei in metabolic diseases, the direct effects of glucose and fat on the hypothalamus, and the influence of the hypothalamus on metabolic diseases. Furthermore, it investigates the role of the hypothalamus in the emergence and progression of metabolic disorders, including obesity and diabetes. Finally, it outlines the current research progress in treating metabolic diseases through hypothalamus regulation. This study could provide a theoretical basis for understanding the pathophysiological mechanisms of metabolic diseases and development of new treatment strategies.

## INTRODUCTION

The global prevalence of metabolic diseases, including obesity, diabetes, and metabolic syndrome, has reached alarming levels, posing significant challenges to public health. Metabolic syndrome encompasses a cluster of pathophysiological abnormalities—such as insulin resistance, hyperglycemia, hyperlipidemia, and hypertension—that collectively elevate the risk of type 2 diabetes and cardiovascular complications (*Tang, Purkayastha & Cai, 2015*). Central to the regulation of energy balance and systemic metabolism is the hypothalamus, a neuroendocrine hub that integrates peripheral signals (*e.g.*, leptin, insulin) and coordinates neural circuits to modulate appetite, energy expenditure, and glucose homeostasis (*Belgardt & Brüning, 2010*; *Jais et al., 2020*). Dysregulation of hypothalamic nuclei, such as the arcuate nucleus (ARC) and paraventricular nucleus (PVN), disrupts these processes, contributing to the pathogenesis of metabolic disorders (*Adlanmerini et al., 2021*; *Han et al., 2021*).

Recent advances in molecular biology and neuroimaging have deepened our understanding of hypothalamic mechanisms, revealing its dual role as both a sensor of metabolic states and a regulator of peripheral organs. For instance, hypothalamic glucose-sensing neurons and neuropeptidergic pathways (*e.g.*, agouti-related protein

Corresponding author
Guoqi Li, 15512468306@163.com

(AgRP)/pro-opiomelanocortin (POMC) circuits) are now recognized as critical mediators of systemic insulin sensitivity and lipid metabolism. Despite this progress, the interplay among hypothalamic dysfunction, genetic predispositions, and environmental factors in metabolic diseases remains incompletely understood.

## Scope of the review

This review synthesizes current research on the hypothalamus's role in metabolic diseases, with a focus on:

(1) Hypothalamic nuclei dynamics: Functional roles of ARC, PVN, ventromedial nucleus (VMH), and dorsomedial nucleus (DMH), in energy balance and their dysregulation in obesity and diabetes.
(2) Peripheral-hypothalamic crosstalk: Mechanisms by which adipose-derived signals (*e.g.*, leptin, adipokines) and gut hormones (*e.g.*, Glucagon-like peptide-1 (GLP-1)) modulate hypothalamic activity.
(3) Therapeutic innovations: Emerging strategies targeting hypothalamic pathways, including GLP-1 receptor agonists, melanocortin 4 receptor (MC4R) modulators, and neuroimaging-guided interventions.
(4) Translational relevance: Insights from genetic obesity syndromes (*e.g.*, Prader-Willi) and clinical trials of hypothalamic-targeted therapies.

This article is intended for researchers and clinicians across disciplines—neuroscience, endocrinology, pharmacology, and genetics—who seek to bridge foundational insights into hypothalamic function with translational applications. By integrating molecular, physiological, and clinical perspectives, this review aims to catalyze interdisciplinary collaboration and inform novel therapeutic approaches for metabolic diseases.

# SURVEY/SEARCH METHODOLOGY

To ensure comprehensive and unbiased coverage of the literature, a systematic approach was employed to identify relevant studies and articles (Fig. 1). The following steps were taken to achieve this:

## Literature search strategy
### Databases and search engines
The following databases were queried (date range: January 2010–April 2024):
    PubMed, Google Scholar, Web of Science, Scopus, PubMed Central, and the Cochrane Library.

### Search terms and boolean logic
A combination of controlled vocabulary (MeSH terms, Emtree) and free-text keywords was used, tailored to each database. Examples include:
    Hypothalamus-related terms: "hypothalamic nuclei," "arcuate nucleus," "neuroendocrine regulation," "leptin-melanocortin pathway."

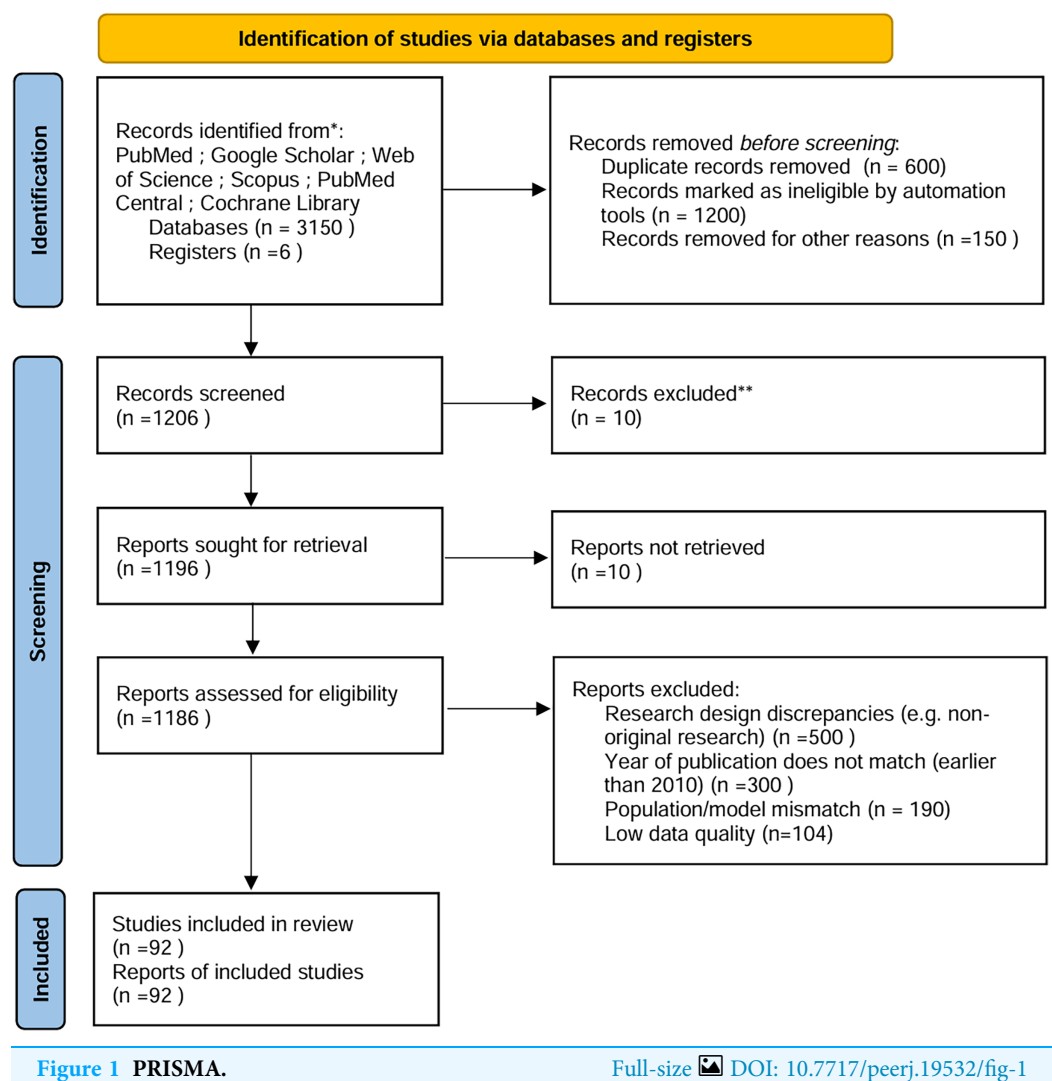

**Figure 1 PRISMA.**                

Metabolic diseases: "obesity," "diabetes mellitus," "monogenic obesity," "Prader-Willi syndrome."

Interventions: "GLP-1 agonists," "setmelanotide," "bariatric surgery (VSG/RYGB)."

Imaging/assays: "PET imaging," "fMRI," "neural circuitry."

Full search syntax for PubMed is provided in Supplemental File S1.

## Inclusion/Exclusion criteria

### Inclusion criteria

**Study types**: Peer-reviewed original research (human/animal), RCTs, systematic reviews, and meta-analyses.

**Population**: Human studies on hypothalamic-linked metabolic disorders (*e.g.*, monogenic obesity, hypothalamic injury) or animal models with translational relevance.

**Interventions**: Pharmacological (*e.g.*, MC4R agonists), surgical (*e.g.*, bariatric surgery), or dietary interventions targeting hypothalamic pathways.

**Outcomes**: Mechanistic data on hypothalamic regulation (*e.g.*, neuroimaging, hormonal assays) or clinically significant metabolic outcomes (*e.g.*, BMI, HbA1c).

**Timeframe**: 2010–2024, to prioritize recent advancements.

### Exclusion criteria

**Study design**: Non-original research (*e.g.*, editorials, opinion pieces), conference abstracts, preprints, studies with <10 participants (human) or inadequate controls (animal).

**Scope**: Articles focused solely on non-hypothalamic pathways (*e.g.*, peripheral insulin signaling without CNS involvement).

**Data quality**: Studies with incomplete methodology, unclear statistical reporting, or high risk of bias (per ROBINS-I/ARRIVE guidelines).

## Screening process and automation tools

### Automation tools

**De-duplication**: EndNote (v20) and Rayyan AI removed 1,248 duplicate records.

**Initial screening**: Rayyan's NLP algorithm prioritized 2,500 records for title/abstract screening based on keyword relevance.

**Exclusions by automation**:

*Pre-screening*: 412 records excluded (reasons: non-English, non-peer-reviewed, or published before 2010).

*Title/abstract filtering*: 1,022 records flagged as low relevance (*e.g.*, non-metabolic hypothalamic topics).

### Manual screening

**Phase 1** Two independent reviewers screened titles/abstracts (κ = 0.86). Discrepancies (*n* = 124) were resolved by a third reviewer.

**Phase 2**: Full-text review of 1,186 articles, with exclusions documented (Fig. 1):

*Research design discrepancies*: 500 excluded (*e.g.*, non-original research).

*Population/model mismatch*: 190 excluded (*e.g.*, non-hypothalamic obesity models).

*Low data quality*: 104 excluded (per predefined quality thresholds).

## Bias mitigation and quality assessment

**Risk of bias**: Assessed using ROBINS-I for human studies and SYRCLE's tool for animal studies.

**Data extraction**: Standardized templates ensured consistency; two reviewers cross-validated data (95% agreement).

**Prioritization**: High-quality RCTs and mechanistic studies with validated endpoints (*e.g.*, fMRI-confirmed hypothalamic activity) were weighted more heavily.

## Transparency enhancements

**PRISMA flowchart**: Expanded to detail automation exclusions, manual screening phases, and reconciliation steps (Fig. 1).

**Exclusion rationale**: Each exclusion category (*e.g.*, "population mismatch") is explicitly showed in Fig. 1.

**Tool validation**: Rayyan's algorithm accuracy was tested against a manually screened subset ($n = 500$); sensitivity = 92%, specificity = 88%.

## Limitations

**Language bias:** Non-English articles ($n = 34$) were excluded but represented <2% of initial results.

**Automation constraints:** NLP tools may overlook nuanced mechanistic data; this was mitigated by manual full-text review.

# PHYSIOLOGICAL FUNCTIONS OF HYPOTHALAMIC NUCLEI

The hypothalamic nuclei play a pivotal role in regulating feeding behavior, energy balance, and metabolism, encompassing structures such as the arcuate, paraventricular, ventromedial, and dorsomedial nuclei. These nuclei contain neurons that modulate crucial physiological processes, including appetite, energy metabolism, and glucose homeostasis, through interactions and connections with other brain regions (Fig. 2).

## Arcuate nucleus

The AgRP and POMC neurons in the ARC are crucial for feeding behavior and weight control. They are regulated by various external signals, including insulin, neurotransmitters, and hormones, which in turn influence feeding behavior, energy expenditure, and glucose metabolism (*Adlanmerini et al., 2021*).

**AgRP neurons**, located within the ARC region of the hypothalamus, play a crucial role in modulating feeding behavior, weight management, and energy metabolism. When the body requires additional energy, such as during hunger, AgRP neurons are activated, leading to increased appetite and reduced energy expenditure, helping the body acquire necessary energy sources. Conversely, when the body stores excess energy, other neurons are activated to inhibit AgRP neuron activity, resulting in decreased appetite and increased energy expenditure. The AgRP neural circuit regulates body weight by influencing the signaling of neurons expressing melanocortin 4 receptor (MC4R) in the dorsal lateral hypothalamic nucleus (*Han et al., 2021*). The activity of AgRP neurons is influenced by various regulatory factors, including neuropeptide Y (NPY). Studies have shown that NPY plays an essential role in the short-term acute effects of AgRP neuron activation. Mice lacking NPY fail to rapidly increase food intake when AgRP neurons are chemically or optogenetically activated (*Engström Ruud et al., 2020*; *Chen et al., 2019*). Additionally, AgRP neuron activation leads to rapid switching of substrate utilization, reducing fat consumption and increasing overall glucose utilization, independent of food intake (*Cavalcanti-de-Albuquerque et al., 2019*). Beyond food intake regulation, AgRP neuron activity plays a critical role in regulating systemic insulin sensitivity and glucose metabolism through distinct neural and endocrine pathways. **1. Acute glucose mobilization:** Optogenetic activation of AgRP neurons rapidly increases hepatic glucose production *via* NPY-dependent sympathetic stimulation of hepatic innervation. This process occurs within minutes, independent of feeding behavior, and requires Y1 receptor (Y1R) signaling in the paraventricular thalamus (PVT) to suppress hepatic insulin receptor

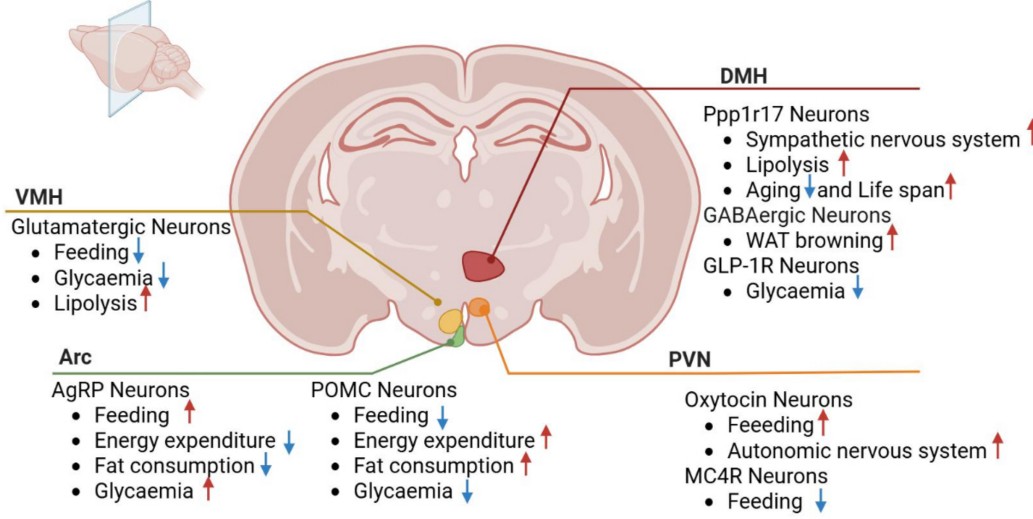

**Figure 2 Roles of hypothalamic nuclei in metabolic diseases.** VMH, ventromedial nucleus; PVN, paraventricular nucleus; Arc, arcuate nucleus; DMH, dorsomedial nucleus; POMC, pro-opiomelano-cortin; GLP-1R, Glucagon-like peptide-1 receptor; Ppp1r17, protein phosphatase 1 regulatory subunit 17. MC4R, Melanocortin-4 Receptor Red arrows depict increase and blue arrows depict decrease.

substrate 2 (IRS2) phosphorylation (*Engström Ruud et al., 2020*). **2. Chronic insulin resistance:** Sustained AgRP hyperactivity induces brown adipose tissue (BAT)-derived myostatin expression, which systemically inhibits insulin signaling in skeletal muscle and white adipose tissue. Genetic ablation of myostatin, specifically in BAT, restores insulin-stimulated glucose uptake, demonstrating a causal role for the AgRP-myostatin axis in metabolic dysregulation (*Steculorum et al., 2016*). **3. Molecular crosstalk:** NPY released from AgRP neurons directly antagonizes leptin signaling in POMC neurons, while myostatin inhibits adiponectin's metabolic effects in peripheral tissues. This dual blockade creates a feedforward loop that exacerbates insulin resistance during prolonged energy surplus (*Engström Ruud et al., 2020*; *Steculorum et al., 2016*). Recent research using CRISPR/Cas9 technology to inactivate the LEPR gene in AgRP neurons of adult mice led to severe obesity and glucose metabolism disorders (*Xu et al., 2018*). The function and activity of AgRP neurons are further modulated by specific vagal afferent types, which transmit signals through the brainstem, exerting moderate temporal effects on AgRP neuron activity related to intestinal chemoreception and mechanoreception (*Bai et al., 2019*). These findings expand our understanding of AgRP neuron regulation, highlighting its dynamic modulation across different time scales. Notably, studies demonstrate that disrupting *TET3*, a key epigenetic regulator involved in DNA demethylation, activates AgRP neurons and triggers adverse metabolic effects such as binge eating, obesity, and diabetes (*Xie et al., 2022*). In AgRP neurons, loss of TET3 disrupts the epigenetic balance of DNA methylation, leading to persistent overactivation of hunger-promoting signaling pathways. In summary, AgRP neurons play a crucial role in regulating feeding, weight, and energy metabolism and are significant for understanding metabolic diseases such as

obesity and diabetes. Therefore, AgRP neurons have become potential therapeutic targets, and related drug development is actively ongoing. These research findings provide important scientific evidence for future treatments of metabolic diseases.

**POMC neurons**, located in the ARC of the hypothalamus, play a crucial role in transmitting signals related to body weight and obesity. These neurons are highly sensitive to external signals, including neurotransmitter changes under hunger conditions, and maintain weight balance by influencing feeding behavior and energy metabolism. The regulatory role of POMC neurons in glucose homeostasis involves intricate neuroanatomical and molecular mechanisms that produce paradoxical effects depending on three key variables: **1. Projection-specific functions:** While global inhibition of POMC neurons reduces blood glucose *via* enhanced hypothalamic insulin signaling (*Üner et al., 2019*), selective activation of their descending projections to the liver paradoxically increases hepatic glucose production through α1-adrenergic receptor-mediated sympathetic stimulation. This suggests that POMC neurons contain functionally antagonistic subpopulations along the neuroaxis (*Üner et al., 2019*; *Kwon et al., 2020*). **2. Receptor subtype dynamics:** The glucoregulatory dichotomy may stem from the differential engagement of melanocortin receptors (MCRs). Hepatic-projecting POMC neurons predominantly signal through the MC4R-αMSH axis to activate cAMP/PKA in liver stellate cells, whereas glucose-lowering effects involve MC3R-dependent potentiation of pancreatic β-cell insulin secretion *via* vagal efferents (*Qi et al., 2023*). **3. Metabolic state dependency:** In obese mice, POMC activation fails to suppress glucose due to leptin resistance-induced downregulation of MC4R in the nucleus tractus solitarius (NTS). This state-specific impairment is rescued by concurrent GLP-1 receptor agonism, revealing a compensatory interaction between melanocortin and incretin systems (*Chen et al., 2022*). In addition to glucose homeostasis control, POMC neurons interact with other types of neurons, such as prepronociceptin and NPY neurons. These neurons jointly regulate feeding behavior and energy metabolism, thereby affecting weight balance (*Chen et al., 2022*). Some studies also suggest an interaction between POMC neurons and astrocytes, which affects energy metabolism and adipocyte function (*Chen et al., 2022*; *Yi et al., 2011*). Furthermore, the important role of steroid receptor coactivator-1 in hypothalamic POMC neurons has been revealed, affecting leptin-regulated feeding and body weight (*Yang et al., 2019*). The communication network within the ARC of the hypothalamus regulates energy expenditure and food intake through lactate signaling. This process involves the conversion of glucose to lactate by elongated cells and its transmission to POMC neurons through monocarboxylic acid transporters (*Lhomme et al., 2021*). Primary cilia are crucial for obesity regulation during the development of hypothalamic neurons. Knockout of the cilia-forming genes IFT88 and KIF3A has been shown to inhibit ciliogenesis in POMC neurons, ultimately affecting obesity status in adult mice (*Lee et al., 2020*). Overall, research on POMC neurons and their related signaling pathways provides new insights into the mechanisms of obesity regulation. These findings help reveal the mechanisms underlying obesity development and provide guidance for developing more effective interventions to manage and prevent obesity-related diseases.

Glucose-sensing cells, particularly specialized neurons in the hypothalamus, play a pivotal role in detecting blood glucose fluctuations and maintaining systemic glucose homeostasis. These neurons are classified into two subtypes: glucose-excited (GE) neurons and glucose-inhibited (GI) neurons. GE neurons are activated by elevated glucose levels through ATP-sensitive potassium (KATP) channels, whereas GI neurons are inhibited by high glucose and rely on chloride channels such as Anoctamin 4 (Ano4) for activation under hypoglycemic conditions (*Tu et al., 2023*; *He et al., 2020*). Notably, in the ventrolateral ventromedial hypothalamus (vlVMH), a subset of GI neurons co-expressing estrogen receptor-α (ERα) and Ano4 has been identified as critical sensors for hypoglycemia, triggering counterregulatory responses (*e.g.*, glucagon release) *via* projections to the arcuate nucleus (ARC) and dorsal raphe nuclei (DRN) (*He et al., 2020*).

Dysfunction in glucose-sensing neurons is a hallmark of metabolic diseases. In obesity and type 2 diabetes, chronic hyperglycemia disrupts GI neuron activity by suppressing AMP-activated protein kinase (AMPK) and enhancing mammalian target of rapamycin (mTOR) signaling, leading to impaired nitric oxide (NO)-mediated insulin sensitivity. Additionally, hypothalamic inflammation and extracellular matrix (ECM) remodeling in obesity induce fibrosis around AgRP neurons, further blunting insulin signaling and promoting hyperphagia. Intriguingly, in type 1 diabetic models, genetic ablation of Ano4 in VMH GI neurons normalizes blood glucose levels, suggesting that targeting Ano4 may offer therapeutic potential (*Tu et al., 2023*; *He et al., 2020*). Simultaneously, SHLP2 (a mitochondrial-derived peptide) participates in appetite and energy metabolism regulation. It modulates neuronal activity in appetite-control circuits (*e.g.*, POMC/NPY neurons), thereby influencing weight homeostasis. The interplay between glucose-sensing neurons and SHLP2 highlights a dual regulatory mechanism linking glucose sensing to energy balance (*Li et al., 2019*).

A recent spatial transcriptomic study in crab-eating macaques revealed distinct hypothalamic responses to obesity and diabetes. The ARC exhibited heightened inflammatory immune activity, while the PVN showed suppressed metabolic activity (*Lei et al., 2024*). This contrast highlights functional interplay between the ARC and PVN in energy homeostasis, suggesting that ARC-driven inflammation may exacerbate PVN dysfunction in metabolic diseases.

Overall, the ARC of the hypothalamus, as a complex regulatory center, plays a pivotal role in energy metabolism and feeding behavior. Further research into the functions and interaction mechanisms of neurons within the ARC of the hypothalamus can help reveal the pathogenesis of obesity and metabolic diseases, providing theoretical support and clinical guidance for future treatment strategies.

## Paraventricular nucleus

The PVN of the hypothalamus serves as a central hub for integrating energy balance signals through anatomically defined connections with specific brain regions and peripheral tissues, and its neuropeptidergic neuronal populations orchestrate both autonomic and behavioral responses to metabolic demands.
### Circuit-specific connectivity of PVN neurons

The PVN establishes bidirectional connections with: **1. Brainstem nuclei:** The nucleus of the solitary tract (NTS) provides catecholaminergic input (*e.g.*, norepinephrine) *via* NTSTH neurons, which relay visceral hunger signals (*e.g.*, hypoglycemia) to PVNMC4R neurons. Optogenetic activation of NTSTH→PVN projections mimics fasting-induced hyperphagia (*Zhang et al., 2020*). The parabrachial complex (PBN) receives glutamatergic projections from PVNMC4R neurons, modulating satiety and aversion responses. These connections are critical for MC4R-dependent suppression of feeding (*Wang et al., 2021*). **2. Hypothalamic regions:** ARC: PVN integrates orexigenic (AgRP) and anorexigenic (POMC) signals through direct synaptic inputs. Notably, 63% of Kiss1r-expressing neurons in the ARC are POMC neurons, suggesting crosstalk between reproductive and metabolic circuits. VMH: Adrenergic PVN neurons modulate VMH-driven sympathetic outflow to white adipose tissue (WAT), linking central MC4R signaling to peripheral lipolysis. **3. Peripheral tissues:** PVNMC4R neurons regulate hepatic glucose production *via* sympathetic innervation of the liver and influence adipokine secretion (*e.g.*, leptin resistance) through projections to WAT (*Sayar-Atasoy et al., 2023*).

### Neuropeptidergic identity of PVN neurons

The PVN contains distinct peptidergic populations with specialized roles: **1. Oxytocin (Oxt) neurons:** Oxt neurons suppress appetite by enhancing ARC POMC neuron activity and inhibiting AgRP/NPY neurons. They also project to the NTS to potentiate vagal satiety signals (*Wang et al., 2023*). Genetic ablation of Oxt receptors in the PVN exacerbates diet-induced obesity, highlighting their role in energy homeostasis (*Xiao et al., 2021*). **2. Corticotropin-releasing hormone (CRH) neurons:** CRH neurons activate the HPA axis under energy deficit, increasing gluconeogenesis and mobilizing lipid reserves. Chronic HFD disrupts CRH-mediated feedback, contributing to metabolic syndrome (*Douglass et al., 2023*). **3. Thyrotropin-releasing hormone (TRH) neurons:** TRH neurons regulate thermogenesis *via* sympathetic activation of brown adipose tissue (BAT). MC4R agonists potentiate TRH release, linking melanocortin signaling to adaptive thermogenesis (*Sayar-Atasoy et al., 2023*).

### MC4R signaling mechanisms in PVN

**1. Ciliary dependency:** MC4R requires primary cilia for proper signaling. Cilia defects impair cAMP/PKA activation in PVN neurons, leading to hyperphagia and reduced energy expenditure (*Sayar-Atasoy et al., 2023*). **2. Adrenergic crosstalk:** Norepinephrine dynamically modulates PVNMC4R activity through α1-ARs, which enhance GABAergic inhibition from ARC AgRP terminals. This mechanism is essential for fasting-induced feeding (*Yang et al., 2022*).

The PVN's role in energy balance hinges on its anatomically discrete connections (NTS, PBN, ARC, liver, WAT) and neuropeptide-specific populations (Oxt, CRH, TRH). These findings refine the mechanistic understanding of PVN circuitry and highlight potential targets for obesity therapies, such as tissue-specific MC4R modulators or Oxt receptor agonists.
## Ventromedial nucleus

The VMH assumes a pivotal position in modulating feeding behaviors and energy metabolism, crucial for elucidating and managing metabolic disorders. The O-linked N-aclcNAc transferase (OGT) within VMH neurons plays a central role in regulating body mass and lipid homeostasis by modulating white adipose tissue (WAT) lipolysis and energy expenditure. Genetic ablation of OGT in mice induces obesity, weight gain, and metabolic dysfunction through disrupted fat mobilization (*Wang et al., 2022*). However, OGT's systemic functions extend far beyond energy balance, necessitating cautious therapeutic targeting to avoid unintended consequences. Key underemphasized roles include: **1. Genomic stability maintenance:** OGT stabilizes translesion DNA polymerase η (Pol η) *via* O-GlcNAcylation at Thr457, ensuring efficient DNA damage repair and preventing replication fork collapse under genotoxic stress (*Sayar-Atasoy et al., 2023*). **2. Transcriptional fidelity control:** By dynamically assembling RNA polymerase II clusters, OGT regulates transcriptional elongation and mRNA splicing, as exemplified by its role in SRPK2-mediated lipid synthesis gene expression in tumors (*Chatham, Zhang & Wende, 2021*; *Nie et al., 2020*). **3. Tumor metabolic reprogramming:** OGT redirects glucose flux into the pentose phosphate pathway (PPP) *via* PFK1 Ser529 O-GlcNAcylation, enhancing NADPH production to support cancer cell survival under oxidative stress (*Nie et al., 2020*). Furthermore, insights into the mechanisms of VMH neurons in regulating food consumption have been unveiled. Specifically, targeted activation of VMH neurons that express steroidogenic factor 1 leads to a swift suppression of food intake, underscoring their pivotal role in appetite control (*Yang et al., 2022*). The cilia of neurons in VMH play an important role in weight regulation. Studies have found that the enrichment of AC3 (type III adenylate cyclase) in cilia is crucial for resisting obesity and affects body weight by regulating autophagy, providing a new perspective on understanding weight regulation mechanisms (*Tu et al., 2023*). Furthermore, the Ano4 channel holds a pivotal position in glucose-sensitive neurons residing within the VMH. Genetic ablation of the *Ano4* gene in mice results in lowered blood glucose levels and compromises the counterregulatory mechanisms triggered during hypoglycemic events. Conversely, activation of Ano4-expressing neurons in the VMH of diabetic mice elevates food consumption and blood glucose concentrations, while their sustained inhibition ameliorates hyperglycemia, suggesting that the Ano4 channel could represent a promising therapeutic avenue for addressing aberrant feeding patterns and glucose dysregulation-associated disorders (*He et al., 2020*). Neurons within the ventrolateral VMH, expressing estrogen receptor alpha (ERα), demonstrate sensitivity to glucose fluctuations. Activation of the ERα-mediated pathway projecting to the medioposterior ARC of the hypothalamus, and concurrent inhibition of the ERα signaling loop targeting the dorsal Raphe nuclei, both elevate blood glucose levels, thereby safeguarding against severe hypoglycemia in mice (*Zhang et al., 2024*). Recent investigations have highlighted the abundant expression of hypothalamic ventromedial secretin in the CNS, underscoring its pivotal role in modulating satiety sensations, energy metabolism, and bone homeostasis. Secretin deficiency in mice is associated with decreased bone mineral density, hyperphagia,

disrupted adipogenesis, and obesity, whereas its overproduction promotes bone accrual (*Tokizane, Brace & Imai, 2024*).

These discoveries not only deepen our understanding of the role of VMH in energy metabolism but also provide potential targets for developing new treatment strategies for obesity and other metabolic diseases.

### Dorsomedial nucleus

The DMH is an important region of the brain located in the dorsomedial part of the hypothalamus. It participates in regulating the body's homeostasis through interactions with other brain regions, such as other hypothalamic nuclei, the brainstem, and the autonomic nervous system. Recent studies have revealed the significant role of this brain region in regulating energy balance, aging rate, and lifespan. The DMH$^{Ppp1r17}$ neurons in the DMH are particularly noteworthy. Activation of these neurons can stimulate the sympathetic nervous system, promoting the release of lipids and eNAMPT from white adipose tissue. This not only provides energy for physical activity but has also been found to delay aging and increase the lifespan of mice (*Papazoglou et al., 2022*).

This provides strong scientific support for the future development of new therapies to treat obesity and metabolic syndrome, delay aging, and increase lifespan.

In summary, in-depth research on the role of hypothalamic nuclei in metabolic diseases and their underlying mechanisms is crucial for understanding the pathogenesis of these diseases, discovering novel therapeutic targets and devising more efficacious treatment strategies.

## THE CENTRAL ROLE OF THE HYPOTHALAMUS IN METABOLIC DISEASES

The hypothalamus, as a key region of the brain, plays a pivotal role in regulating the body's metabolic balance. It influences energy metabolism, appetite regulation, and glucose homeostasis (Fig. 3) and maintains metabolic health through complex neural circuits closely connected with peripheral metabolic organs (Fig. 4).

### Interaction between the hypothalamus and peripheral metabolic organs

The hypothalamus precisely regulates insulin secretion in the pancreas through diverse neuronal subpopulations, such as oxytocin neurons originating from the PVN. This neural circuit reaches pancreatic β-cells *via* the sympathetic autonomic branch, and activating or silencing of these neurons can lead to hyperglycemia or hypoglycemia, respectively, ensuring stable blood glucose levels (*Wang et al., 2023*). Moreover, the hypothalamus influences body weight and fat metabolism by regulating fat tissue lipolysis and browning processes. For instance, the RIIβ subunit of cAMP-dependent protein kinase A (PKA) in GABAergic neurons within the DMH region modulates the browning of white adipose tissue (WAT). In wild-type DMH GABAergic neurons, the RIIβ subunit combines with the C subunit to form type II PKA, maintaining normal PKA activity and phosphorylation levels of gamma-aminobutyric acid (GABA) receptors, thereby sustaining appropriate

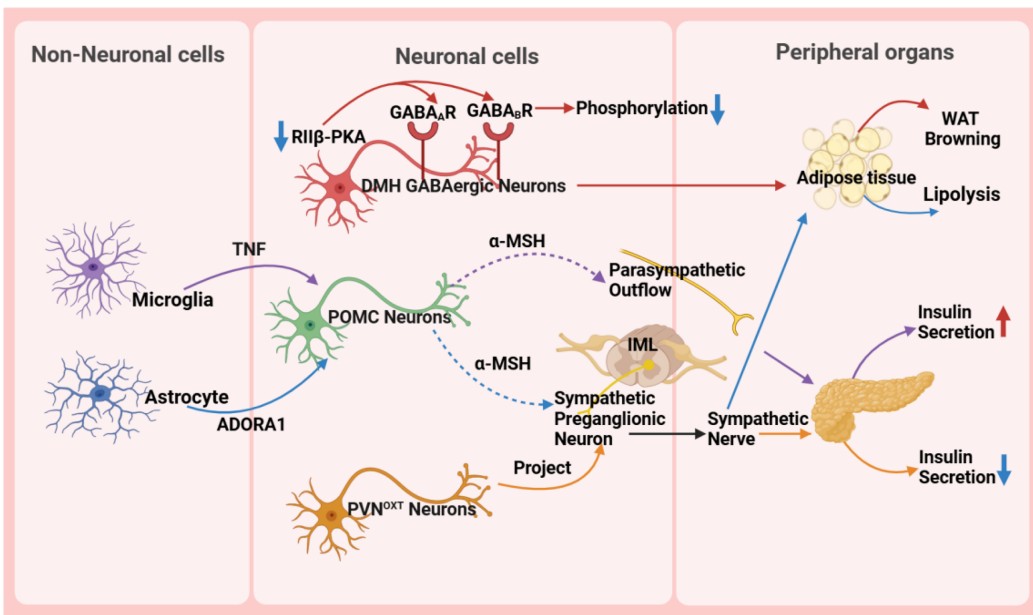

**Figure 3 Role of the hypothalamus in metabolic disease.** GABA $_A$ R, gamma-aminobutyric acid type A receptor; GABA $_B$ R, gamma-aminobutyric acid type B receptor; RIIβ-PKA, the RIIβ subunit of protein kinase A; WAT, white adipose tissue; TNF, tumor necrosis factor; ADORA1, adenosine A1 receptor; DMH, dorsomedial nucleus; α-MSH, α-melanocyte-stimulating hormone; IML, intermediolateral nucleus of the spinal cord; PVN OXT , oxytocin neurons in the paraventricular nucleus.

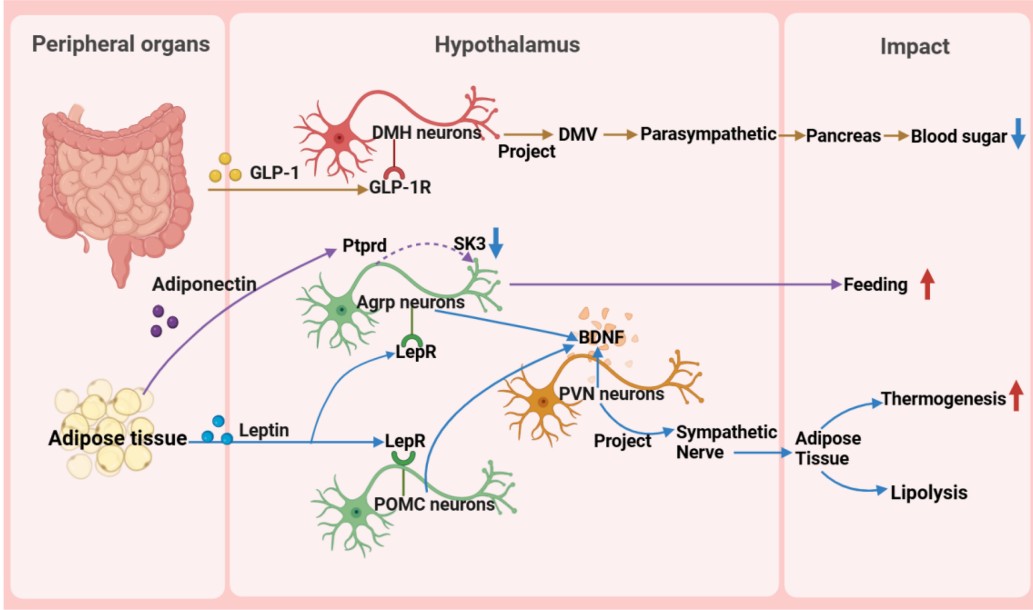

**Figure 4 Reverse impact of peripheral organs on the hypothalamus.** BDNF, brain-derived neuro-trophic factor; DMH, dorsomedial nucleus of the hypothalamus; DMV, dorsal motor nucleus of the vagus nerve; GLP-1, glucagon-like peptide-1; GLP-1R, glucagon-like peptide-1 receptor; POMC, pro-opiomelanocortin; SK, small-conductance calcium-activated potassium channels.

neuronal excitability. However, in neurons with RIIβ mutations, the absence of RIIβ leads to compensatory increases in the RIα subunit and reduced PKA activity. Concurrently, subcellular relocalization of PKA results in inadequate phosphorylation and desensitization of GABA receptors, including GABA type A receptors and GABA type B receptors, on dendrites, thereby increasing neuronal excitability. This heightened excitability, *via* enhanced sympathetic signals, promotes WAT browning, ultimately resulting in a lean phenotype in RIIβ-KO mice. This provides new insights into clinically promoting WAT browning and treating obesity and other metabolic disorders (*Douglass et al., 2023*).

Non-neuronal cells, such as microglia and astrocytes, also play crucial roles in metabolic regulation in the hypothalamus. Microglia, an essential type of glial cell in the nervous system, holds a pivotal position in metabolic regulation. They are important for glucose homeostasis regulation, and recent studies suggest a tumor necrosis factor α (TNF-α)-dependent mechanism. This mechanism can increase the secretion of POMC-derived α-melanocyte-stimulating hormone (α-MSH) and activate other glucose-sensing neurons in the hypothalamus, ultimately leading to a significant increase in insulin secretion through the parasympathetic nerve pathway (*Kim et al., 2019*). A high-fat diet (HFD) activates hypothalamic microglia, triggering inflammatory responses that damage neurons and disrupt glucose and lipid metabolism, ultimately promoting obesity. Uncoupling protein 2 (UCP2) is a critical regulator in this process. HFD upregulates UCP2 expression in microglia, which enhances NLRP3 inflammasome activation and induces the release of key pro-inflammatory cytokines, including TNF-α, IL-1β, IL-6, and the chemokine MCP-1. These cytokines impair the activity of POMC neurons in the arcuate nucleus, leading to hyperphagia and weight gain. Notably, microglia-specific UCP2 knockout mitigates HFD-induced neuroinflammation by suppressing these cytokine cascades, thereby restoring POMC neuron function and reducing obesity (*Rahman et al., 2020*). These findings highlight the importance of microglia in metabolic regulation and disease treatment, providing a new perspective for understanding the role of glial cells in the nervous system.

Astrocytes are involved in the diabetic phenotype and influence the AMPK signaling pathway and neuropeptide circuits that regulate feeding behavior through the modulation of the pyruvate dehydrogenase kinase (PDK2)-lactate axis, leading to metabolic imbalance and hypothalamic inflammation. Inhibiting PDK2 in astrocytes can alleviate diabetes-induced hypothalamic inflammation and alterations in feeding patterns (*Bentsen et al., 2020*). Furthermore, astrocytes are involved in the treatment process of type 2 diabetes. Injection of fibroblast growth factor 1 (FGF1) can improve hyperglycemia, with the main target being glial cells in the mediobasal hypothalamus. Further research has shown that Agrp neuron transmission is necessary for the sustained anti-diabetic effect of FGF1. This discovery reveals the potential mechanism of microglia in diabetes treatment (*Herrera Moro Chao et al., 2022*). In the PVN, chemogenetic manipulation of astrocytes bidirectionally controls the activity of neighboring neurons, autonomic outflow, glucose metabolism, and energy balance by regulating ambient glutamate levels (*Zhang et al., 2024*). Some studies also indicate that astrocytes play a crucial role in regulating the

sympathetic activity and function of adipose tissue (*Chen et al., 2022*; *Yi et al., 2011*). The research results indicate that astrocytes activate POMC neurons located in the ARC through the adenosine A1 receptor signaling pathway. Upon activation, POMC neurons synthesize and release neuropeptides such as α-MSH. Subsequently, α-MSH acts on the sympathetic preganglionic neurons in the intermediolateral nucleus of the spinal cord, which serve as crucial nodes in regulating adipose tissue metabolism. By activating these sympathetic preganglionic neurons, α-MSH promotes the release of neurotransmitters such as NE, which then bind to β-adrenergic receptors on adipocytes, triggering lipolysis in these cells (*Chen et al., 2022*).

The hypothalamus secretes various neurotransmitters, such as oxytocin and orexin, forming a complex network that regulates fat metabolism and whole-body glucose homeostasis. Oxytocin accelerates fat consumption by promoting the breakdown of fat cells (*Li et al., 2024*; *Xiao et al., 2021*). Notably, the hypothalamus plays a crucial role in regulating whole-body glucose homeostasis through its secretion of orexin. Orexin receptors type 1 and 2 are expressed in the DMH and dorsolateral nucleus, respectively, and act together on peripheral metabolic organs such as brown adipose tissue, skeletal muscle, and the liver. Orexin receptor type 1 promotes glucose utilization in peripheral tissues, while receptor type 2 inhibits hepatic gluconeogenesis. These two receptors work synergistically to maintain glucose homeostasis. This complex regulatory network highlights the key role of the hypothalamus in energy balance and metabolic regulation (*Schwartz & Porte, 2005*).

In summary, the complex interactions of the hypothalamus and its internal cell types constitute the core hub of metabolic disease regulation. An in-depth understanding of these regulatory mechanisms not only enhances comprehension of the maintenance and imbalance of metabolic balance but also provides new avenues for treating and preventing related diseases.

## Peripheral organs' influence on the hypothalamus

Peripheral metabolic organs exert an impact on the hypothalamus, particularly those involved in glucose and fat metabolism. These effects manifest not only in the regulation of energy metabolism but also in food intake, weight control, neuronal activity, and metabolic processes.

First, in glucose metabolism, fluctuations in blood glucose levels directly trigger the hypothalamic response mechanism (*Huang et al., 2022*). Glucagon-like peptide-1 (GLP-1) from the intestine precisely activates target neurons in the DMH through its specific receptors, utilizing a mechanism of cAMP-PKA-dependent potassium current inhibition. These activated neurons subsequently regulate the dorsal motor nucleus of the vagus nerve, directly influencing the pancreas through the parasympathetic pathway to finely modulate insulin secretion (*Zhang et al., 2021*). Additionally, the exploration of the glucose-dependent insulinotropic polypeptide receptor signaling pathway has revealed its central role in energy metabolism regulation and provided a theoretical basis for developing innovative drugs for metabolic diseases, such as GLP-1 and glucose-dependent insulinotropic polypeptide dual agonists (*Rupp et al., 2023*). Notably, the newly discovered

GABAergic LepRb$^{Glp1r}$ neuronal population, under the synergistic action of leptin and GLP-1R agonists, exhibits potent feeding suppression and weight management efficacy, further unveiling the exquisite regulatory network of the central nervous system in energy balance (*Zhang et al., 2023*).

## Adipokines and hypothalamic metabolic regulation

Fat metabolism is closely linked to the hypothalamus. Molecular signals derived from fats are transmitted to the hypothalamus through the blood circulation system, affecting neuronal activity and metabolic processes. For instance, the role of acyl-CoA binding protein in astrocytes reveals the importance of fats in energy balance control (*Yi et al., 2011*). Additionally, fat-derived neuregulin 4 acts on the ErbB4 receptor in the hypothalamus, further regulating feeding behavior and weight gain (*Li et al., 2019*).

Lipin and leptin are two important lipid factors that play crucial roles in regulating hypothalamic nuclear function and metabolic activity. Lipin promotes glucose production in the liver and activates the cAMP signaling pathway by binding to the olfactory receptor OLFR734, thereby stimulating hypothalamic appetite (*Feng et al., 2023*). Lipin stimulates feeding by activating AgRP neurons in the arcuate nucleus, a process dependent on specific potassium channels. Pharmacological blockade of this pathway reduces hyperphagia, highlighting its therapeutic potential for obesity (*Butiaeva et al., 2021*). In summary, lipin plays a vital role in regulating metabolism and appetite, and a deeper understanding of its mechanism may provide new directions and strategies for treating metabolic diseases.

Leptin plays a pivotal function in regulating hunger sensations and maintaining energy equilibrium, facilitated by its receptor. Pericytes, a fundamental constituent of the blood-brain barrier, facilitate the passage of circulating leptin into the hypothalamus by virtue of their leptin receptor expression. Upon entry into the hypothalamus, leptin exerts its influence by inhibiting the activity of neurons that stimulate appetite, while potentially enhancing the functionality of other interconnected neurons through the activation of diverse signaling cascades. This orchestrated regulation subsequently modulates energy intake and expenditure processes (*Torres & Pitts, 2021*; *Jiang et al., 2020*). Studies have revealed that Sh2b1 in leptin receptor neurons plays a crucial role in regulating the sympathetic nervous system and brown adipose tissue function and is an important component for maintaining the normal operation of the SNS/BAT/thermogenesis axis. By affecting this axis, Sh2b1 plays a significant role in body temperature regulation and energy metabolism, as well as obesity and metabolic diseases. Research results indicate that the deletion or dysfunction of Sh2b1 leads to cold intolerance, obesity, insulin resistance, and hepatic steatosis, whereas overexpressing Sh2b1 has a protective effect. This discovery provides a new potential target for the treatment of obesity and its complications (*Wang et al., 2020*). Moreover, research has revealed metabolic disorders caused by mutations in the leptin gene (ob), particularly obesity, thermogenesis, and lipolysis defects, which are closely related to the reduced sympathetic innervation of adipose tissue. Chronic leptin treatment can restore sympathetic innervation in adipose tissue of ob/ob mice, thereby correcting functional defects. This restoration process is orchestrated by AgRP and POMC neurons within the ARC region of the hypothalamus, which express leptin receptors, and

the signaling cascade is propagated *via* brain-derived neurotrophic factor-expressing neurons located in the PVN. This underscores the significance of leptin signaling in modulating the plasticity of sympathetic innervation in adipose tissue through intricate neural pathways, a vital mechanism for preserving energy balance (*Elmaleh-Sachs et al., 2023*). These findings provide important insights into understanding the pathogenesis of obesity and related metabolic diseases and offer guidance and inspiration for the development of drugs and therapies for obesity treatment in the future.

In summary, the influence of peripheral organs on the hypothalamus is multifaceted, involving not only the regulation of glucose and fat metabolism but also neuronal activity and metabolic processes. Together, these influences form a complex network that ensures the balance and stability of energy metabolism in the body. Delving deeper into these mechanisms can enhance understanding of the pathogenesis of obesity and related metabolic diseases and provide important guidance for developing new therapeutic drugs and treatments.

# HYPOTHALAMIC DYSFUNCTION AND GENETIC OBESITY

## Monogenic obesity: clinical relevance of MC4R and melanocortin pathways

Loss-of-function mutations in the MC4R gene (*e.g.*, frameshift variant c.732_735delCAGT) disrupt hypothalamic melanocortin-4 receptor signaling, leading to dominantly inherited early-onset obesity (*Correa-da-Silva et al., 2024*). Clinical studies reveal that carriers of pathogenic MC4R mutations exhibit leptin resistance, elevated insulin levels, and skeletal muscle metabolic abnormalities (*Bochukova et al., 2018*). Notably, the obesity phenotype in these individuals is modulated by polygenic risk scores (PRS), with high-PRS carriers showing an additional BMI increase of 2.1 kg/m$^2$ (*Chami et al., 2020*). Common variants near MC4R (*e.g.*, rs17782313) are also linked to abdominal obesity and insulin resistance, reinforcing the pathway's role in polygenic obesity (*Yeo et al., 1998*).

## Syndromic obesity: hypothalamic pathophysiology in Prader-Willi syndrome

Prader-Willi syndrome (PWS), caused by paternal deletions or imprinting defects in the 15q11.2-q13 region, involves hypothalamic neuron-glia network dysfunction (*Vaisse et al., 1998*). Patients with larger deletions (T1 subtype) exhibit microglial phagolysosomal defects in the hypothalamus, impairing neuronal debris clearance and disrupting AQP4-mediated glymphatic drainage, which exacerbates neuroinflammation and hyperphagia (*Krude et al., 1998*). Transcriptomic analyses of PWS hypothalamic tissue show aberrant activation of AgRP/NPY pathways during fasting, alongside reduced BDNF expression, disrupting energy balance (*Marenne et al., 2020*). Additionally, PHIP mutations repress POMC transcription, linking genetic defects to leptin signaling impairment (*Asai et al., 2013*). These studies underscore the pivotal role of hypothalamic neuron-glia crosstalk in genetic obesity and unravel the complex interplay between monogenic and polygenic mechanisms. Future research must prioritize integrating human multi-omics data (*e.g.*, single-cell transcriptomics and epigenetics) with functional validation in organoid models

to accelerate therapeutic discovery. By incorporating translational evidence—such as DCCR trial outcomes and MC4R-PRS interaction models—this review bridges foundational insights to clinical practice, aligning with the medical community's demand for actionable knowledge.

# NEUROIMAGING AND INTERVENTIONS FOR HYPOTHALAMIC METABOLISM

## Application of neuroimaging techniques in hypothalamic metabolic disorders

As core functional neuroimaging technologies, positron emission tomography (PET) and single-photon emission computed tomography (SPECT) provide essential tools for investigating the pathological mechanisms underlying hypothalamic metabolic dysregulation (*Chung et al., 2025*). Clinical studies have revealed that obese patients exhibit significantly reduced glucose metabolic rates in the hypothalamus and associated brain regions (*e.g.*, striatum, prefrontal cortex), a phenomenon strongly correlated with enhanced insulin resistance and impaired leptin signal transduction (*Al-Alsheikh et al., 2023*).

Recent studies utilizing the radiolabeled glucose tracer (18F-FDG) in combination with high-resolution PET imaging have uncovered dynamic changes in glucose metabolism patterns during brain maturation in adolescent rats. Experimental results demonstrate that glucose metabolism becomes markedly enhanced in specific brain regions (*e.g.*, hippocampus, thalamocortical circuits) as the nervous system matures, with metabolic hotspots exhibiting precise spatiotemporal synchronization with the activation of long-range neural connections (*e.g.*, cortico-limbic pathways). Further analyses indicate that the dynamic reorganization of metabolic networks optimizes energy allocation efficiency for synaptic transmission, thereby promoting the formation of functional neural circuits (*Choi et al., 2015*). These findings provide critical theoretical insights into the potential impact of metabolic disorders (*e.g.*, obesity, diabetes) on adolescent brain development: aberrant metabolic signaling may disrupt the energy supply to neural networks, thereby impairing the development of synaptic plasticity and cognitive functions.

# INTERVENTION STRATEGIES FOR HYPOTHALAMIC METABOLIC DISORDERS

## Drugs targeting the hypothalamus for metabolic diseases treatment

Metabolic disorders like obesity and T2D are closely linked to the hypothalamus, a vital regulator of hunger sensations, energy usage, and glucose balance within the body. In recent years, researchers have developed drugs targeting the hypothalamus to treat these metabolic diseases (Table 1).

GLP-1 receptor agonists mimic the physiological actions of GLP-1. After eating, GLP-1 acts on the hypothalamus, suppressing appetite, delaying gastric emptying, increasing glucose-dependent insulin release, reducing glucagon secretion, and promoting the growth

**Table 1 Drug categories and representative drugs with their mechanisms of action.**

| Drug name | Target | Clinical trial | Trial design | Number of participants | Primary outcome |
|---|---|---|---|---|---|
| **Semaglutide** | GLP-1R | STEP Trial | Randomized, double-blind, parallel trial for weekly subcutaneous injections in obese adults (BMI ≥ 30 or ≥27 with comorbidity), excluding diabetics. | STEP 1: 1,961 STEP 3: 611 STEP 4: 902 | Avg. weight loss at 68 weeks: STEP 1-14.9%, STEP 3-16.0%, STEP 4-~6.9% rebound. |
| **Liraglutide** | | RCT | Randomized, controlled, double-blind trial of obese adults stratified by gender & age, evaluating 1-year weight loss maintenance post-diet with exercise, liraglutide, combo therapy, or placebo. | 130 | 1-year sustained weight loss & maintained postprandial satiety scores post-weight loss. |
| **Setmelanotide** | MC4R | Phase 3 Clinical Trial | Multicenter, randomized, double-blind, placebo-controlled trial in BBS/Alström syndrome patients aged ≥6 with obesity, undergoing 16-week daily subcutaneous injection, followed by 52 weeks of open-label setmelanotide. | 38 | 32.3% of BBS patients achieved ≥10% weight loss at 52 weeks. |
| | | Phase 2 Clinical Trial | Open-label, multicenter trial for 6–40-years-old obese patients with hypothalamic injury/disease post-surgery, chemo, or radiation, undergoing 16-week daily subcutaneous injection. | 18 | 89% patients had ≥5% BMI reduction at 16 weeks, averaging 15% decrease from baseline. |

of pancreatic β-cells (*Grunvald et al., 2022*). Representative drugs include liraglutide and semaglutide. In 2021, semaglutide garnered FDA approval in the United States for the management of obesity, administered *via* subcutaneous injection on a weekly basis (*Wilding et al., 2021*). The STEP trial evaluated its efficacy and showed that in obese individuals without diabetes, Specifically, after 68 weeks of treatment, the STEP 1 and STEP 3 trials documented mean weight losses of 14.9% and 16.0%, respectively, far surpassing those observed in the placebo-treated cohorts (*Wadden et al., 2021*; *Rubino et al., 2021*). However, weight significantly rebounds after discontinuation, indicating the importance of long-term use. The STEP 4 trial further confirmed weight rebound after discontinuation, with an average rebound of about 6.9% of weight loss (*Wilding et al., 2022*; *Grunvald et al., 2022*).

In 2014, the Food and Drug Administration (FDA) granted authorization for liraglutide's utilization in managing obesity (*Rubino et al., 2021*). A randomized, controlled, double-blind trial further confirmed the superiority of liraglutide. The study showed that compared with the placebo group, the liraglutide group continued to lose weight 1 year after weight reduction and effectively maintained postprandial appetite suppression and reduced sedentary time, thereby preventing significant weight rebound. Furthermore, when liraglutide was combined with lifestyle interventions such as exercise, the effect was even more significant, further demonstrating the great potential of comprehensive treatment in the management of metabolic diseases (*Jensen et al., 2022*). A meta-analysis indicated that both GLP-1 receptor agonists could reduce the risk of cardiovascular disease events in overweight or obese adults who did not have diabetes (*Leite et al., 2022*). In the SELECT study, semaglutide significantly reduced the risk of cardiovascular events in overweight or obese adults without diabetes (*Lincoff et al., 2023*). Further, when applied to a cohort comprising 529 patients with heart failure and preserved

ejection fraction,semaglutide reduced symptoms related to heart failure and improved physical activity limitations compared with placebo (*Kosiborod et al., 2023*).

These drugs provide new options for the treatment of metabolic diseases by targeting specific mechanisms in the hypothalamus and have shown good efficacy and some cardiovascular protective effects. However, long-term use and weight management after discontinuation still require further research and attention.

MC4R, as mentioned earlier, plays a key role in regulating energy metabolism and appetite in the hypothalamus. Its mechanisms of action include regulating appetite and increasing energy expenditure, receiving metabolic signals to regulate hunger and satiety, enhancing basal metabolic rate, and promoting fat oxidation (*Wang et al., 2021*; *Sayar-Atasoy et al., 2023*). Setmelanotide, as a selective agonist of MC4R, provides a new treatment option for metabolic diseases such as Bardet-Biedl syndrome, Alström syndrome, and hypothalamic obesity.

The clinical research on setmelanotide is extensive and thorough, as exemplified by a Phase 3 clinical trial targeting patients with Bardet-Biedl Syndrome (BBS) and Alström Syndrome. This Phase 3 clinical trial, featuring a multicenter, randomized, double-blinded, and placebo-guided design, encompassed 12 prestigious research institutions situated across the United States, Canada, the United Kingdom, France, and Spain, involving a total of 38 eligible patients with obesity. The trial results showed that after 52 weeks of setmelanotide treatment, 32.3% of patients with BBS aged ≥12 years achieved at least 10% reduction in body weight, a statistically significant result ($p = 0.0006$). Although the effect in patients with Alström syndrome remains uncertain, this finding provides a new therapeutic option for obesity in patients with BBS (*Haqq et al., 2022*). Setmelanotide has exhibited promising therapeutic potential in managing hypothalamic obesity, a condition marked by rapid and excessive weight accrual subsequent to hypothalamic insult. A pivotal Phase 2, open-access, multicenter clinical study involving subjects with hypothalamic obesity observed the administration of setmelanotide to 18 participants, spanning an age range from 6 to 40 years. These individuals had either a documented history of hypothalamic impairment or were diagnosed with benign tumors impacting the hypothalamus, necessitating surgical intervention, chemotherapy, or radiation therapy. Notably, the study outcomes revealed that a remarkable 89% of the cohort achieved the primary end-point metric at week 16, signifying a minimum 5% reduction in their baseline Body Mass Index (BMI). The average BMI reduction for all patients was 15%, indicating that setmelanotide has significant efficacy in treating hypothalamic obesity (*Roth et al., 2024*). The clinical research achievements of setmelanotide have introduced new prospects in the field of metabolic disease treatment, and its precision treatment strategy based on the MC4R mechanism paves the way for future drug development.

Collectively, these medications exert their therapeutic effects by modulating diverse signaling cascades within the hypothalamus, thereby orchestrating appetite regulation, energy expenditure modulation, and glucose homeostasis in the body. This complex interplay ultimately contributes to the effective management of metabolic disorders, including obesity and type 2 diabetes mellitus. With a more in-depth understanding of hypothalamic function and metabolic regulation mechanisms, more effective treatment

methods are expected to be developed in the future, providing patients with greater prospects and additional options.

## Bariatric surgery

Vertical sleeve gastrectomy (VSG) and Roux-en-Y gastric bypass (RYGB) are currently the most commonly performed bariatric surgeries worldwide (*doval & Patti, 2023*).

Bariatric surgery remodels the hypothalamic metabolic regulation network through multidimensional mechanisms, ameliorating obesity-related hormonal imbalances and energy-sensing dysfunction. Studies have demonstrated that procedures such as sleeve gastrectomy (SG) and RYGB significantly reduce the secretion of the orexigenic hormone ghrelin while increasing levels of gut-derived hormones glucagon-like peptide-1 (GLP-1) and peptide YY (PYY) (*Sumithran et al., 2011*; *Suzuki, Jayasena & Bloom, 2012*). This hormonal reprogramming directly targets the ARC of the hypothalamus: GLP-1 enhances satiety signals by activating POMC neurons, while reduced ghrelin suppresses the activity of AgRP neurons, synergistically dampening appetite drive. Additionally, postoperative weight loss restores hypothalamic leptin sensitivity, augmenting energy expenditure and suppressing feeding behavior *via* the JAK-STAT pathway. Enhanced vagal afferent signaling is a critical mediator of surgery-induced hypothalamic modulation. Postoperative alterations in gastrointestinal anatomy (*e.g.*, reduced gastric capacity or rapid nutrient delivery to the distal small intestine) activate vagal signaling through mechanical and chemical stimuli, further inhibiting appetite-associated neuronal activity in the lateral hypothalamic area (LHA) (*Miras & le Roux, 2013*).

Post-bariatric surgery, the patient's habenula volume and functional connectivity with the hypothalamus are enhanced, inhibiting negative emotion-driven feeding behavior. Combined fMRI and PET analyses suggest that dopamine signal remodeling in the hypothalamus-striatum pathway is crucial for long-term weight loss. This remodeling of dopamine signaling, by regulating the reward system and satiety, further consolidates post-surgical appetite suppression and improvements in energy metabolism. Through these multi-level mechanisms, bariatric surgery not only facilitates rapid weight reduction in the short term but also maintains long-term metabolic health and reduces the incidence of obesity-related complications (*Wang et al., 2024*).

## Extreme dietary restrictions (Table 2)

**Ketogenic diet:** The ketogenic diet, characterized by extremely low carbohydrate intake (5–10%) and high fat consumption (70–80%), induces a state of ketosis in which ketone bodies (*e.g.*, β-hydroxybutyrate (BHB)) replace glucose as the primary energy source. Recent studies reveal that the ketogenic diet not only directly suppresses appetite through ketones but also activates synergistic interactions between the gut microbiota and bile acid metabolism. For instance, BHB can bind to amino acids *via* the enzyme CNDP2, forming BHB-amino acid conjugates (*e.g.*, BHB-Phe), which independently inhibit appetite and reduce weight gain in high-fat diet-fed mice, bypassing traditional feeding regulation pathways (*e.g.*, hypothalamic corticotropin or GLP-1 receptor signaling) (*Moya-Garzon et al., 2025*). Additionally, the ketogenic diet upregulates bile acids such as TUDCA and

**Table 2 Extreme dietary patterns: ketogenic diet, very low-carb diet, high-protein diet, and intermittent fasting.**

| Dietary pattern | Key mechanisms |
|---|---|
| **Ketogenic diet** | Induces ketosis (BHB as primary energy) |
| | Upregulates bile acids (TUDCA/TDCA) |
| | Gut microbiota-BHB conjugates suppress appetite |
| **Very low-carb diet** | Mimics fasting state |
| | Depletes glycogen stores—Reduces insulin secretion |
| **Setmelanotide** | Enhances satiety *via* mTORC1 |
| | Increases diet-induced thermogenesis |
| | Downregulates SNAT2 transporters |
| **Intermittent fasting** | Activates hepatic GCN2 |
| | Repairs leptin signaling |
| | Induces "metabolic memory" *via* epigenetic modifications |

Note:
BHB, Beta-hydroxybutyrate; TUDCA, Tauro-ursodeoxycholic acid; SNAT2, Sodium-coupled neutral amino acid transporter 2; GCN2, General Control Nonderepressible 2 (a stress-sensing kinase).

TDCA, reducing intestinal lipid absorption and improving obesity and insulin resistance. This mechanism correlates inversely with BMI and fasting glucose levels in human plasma (*Li et al., 2024*). However, long-term adherence to the ketogenic diet may accelerate cellular senescence in organs, increase cardiovascular risks, and lead to nutrient deficiencies (*Acuña-Catalán et al., 2024*), necessitating professional guidance for safe implementation.

**Very low-carb diet:** Very low-carb diets (daily carbohydrate intake <50 g) mimic a fasting state by reducing insulin secretion and promoting lipolysis. While effective for rapid weight loss and short-term improvement in insulin sensitivity, the adaptation phase often triggers adverse effects such as hypoglycemia, constipation, and dehydration. Research indicates that early-stage carbohydrate restriction depletes glycogen stores, leading to water loss, while insufficient dietary fiber intake disrupts gut function. Long-term application may result in nutrient imbalances (*e.g.*, deficiencies in B vitamins and minerals like potassium and magnesium) and menstrual irregularities in females. Furthermore, prolonged very low-carb diets may dysregulate circadian rhythms, causing nocturnal ghrelin surges and exacerbating metabolic compensatory adaptations (*Acuña-Catalán et al., 2024*). Gradual carbohydrate reduction, prioritizing low-glycemic-index (GI) whole grains and vegetables, alongside electrolyte supplementation, is recommended to mitigate side effects.

**High-protein diet:** High-protein diets (protein >30% of total intake) promote weight loss by enhancing satiety and increasing diet-induced thermogenesis (approximately 20% of caloric expenditure for protein metabolism). However, studies from the University of Washington demonstrate that excessive protein intake (particularly animal protein) may counteract a key metabolic benefit of weight loss—improved insulin sensitivity. In obese females, high-protein diets failed to enhance insulin sensitivity despite weight reduction, whereas moderate-protein diets showed significant improvements. Mechanistically,

high-protein diets may suppress appetite *via* hypothalamic mTORC1 signaling; however, excessive branched-chain amino acids (BCAAs) can disrupt amino acid sensing, triggering compensatory hyperphagia (*Acuña-Catalán et al., 2024*). Long-term high-protein diets may also strain renal function and downregulate intestinal amino acid transporters (*e.g.*, SNAT2), worsening leptin resistance. Protein intake should thus be limited to ≤1.5 g/kg body weight daily, with plant-based proteins prioritized to balance metabolic risks.

**Intermittent fasting:** Intermittent fasting (*e.g.*, 16:8 or 5:2 regimens) activates metabolic repair mechanisms by extending fasting windows, thereby improving insulin sensitivity and energy metabolism. Studies report that alternate-day fasting (ADF) reduces the HOMA-IR index by 53% in insulin-resistant patients, outperforming traditional calorie restriction (*Yin et al., 2022*). Mechanistically, fasting reduces the frequency of insulin secretion, repairs hypothalamic leptin signaling, and enhances hepatic GCN2 expression. This induces epigenetic modifications that establish a "metabolic memory," sustaining insulin sensitivity improvements for months post-intervention (*Yang et al., 2024*). Overall, intermittent fasting represents a sustainable metabolic strategy but requires integration with balanced nutrition and regular exercise to maximize health benefits.

## FUTURE RESEARCH DIRECTIONS

*In-depth study of neuroendocrine signaling pathways*: Future research needs to further explore the mechanisms of neuroendocrine signaling pathways in the hypothalamus, such as the roles of insulin, leptin, and others in the hypothalamus, as well as how they affect physiological processes such as feeding, energy expenditure, and glucose metabolism. This will aid in the comprehension of the pathogenesis of metabolic diseases.

*Elucidation of interactions between hypothalamic nuclei*: The hypothalamus comprises multiple nuclei that interact through intricate neural circuits to regulate metabolic homeostasis. Future research should focus on the mechanisms of interaction between these nuclei and their specific roles in metabolic diseases.

*Precision-based treatment strategies targeting the hypothalamus*: With a deeper understanding of hypothalamic function, precision-based treatment strategies targeting specific neurons or signaling pathways in the hypothalamus can be developed for metabolic diseases such as obesity and diabetes. This will provide new avenues and methods for the treatment of metabolic diseases.

*Integration of interdisciplinary research*: The study of hypothalamic function involves multiple disciplines, such as neuroscience, endocrinology, and genetics. Future research should enhance the integration of and collaboration between these disciplines to jointly promote the in-depth development of hypothalamic function research.

## CONCLUSION

As a neuroendocrine regulatory center, the hypothalamus plays a crucial role in the occurrence and progression of metabolic diseases. Through in-depth research on hypothalamic function, the pathogenesis of metabolic diseases can be better understood, providing new ideas and methods for their treatment. With continuous advancements in molecular biology and imaging technology, future research is expected to achieve more

breakthroughs in hypothalamic function research, contributing significantly to the prevention and treatment of metabolic diseases. Additionally, enhancing the integration and collaboration of interdisciplinary research will provide more possibilities for the in-depth development of hypothalamic function research.

## ACKNOWLEDGEMENTS

I extend my deepest gratitude to Professor Yulin Li for her invaluable conceptual guidance and rigorous editorial revisions, which fundamentally shaped the intellectual framework and scholarly rigor of this work. Her mentorship throughout the research trajectory has been indispensable. I also sincerely thank Dr. Guoqi Li for his critical feedback during manuscript refinement, particularly his expertise in enhancing methodological clarity.

### Funding

This study was supported by the Beijing Hospitals Authority's Ascent Plan (Code: DFL20240603). Additionally, this work received funding from the National Natural Science Foundation of China Youth Program (Grant No. 82100397). The funders had no role in study design, data collection and analysis, decision to publish, or preparation of the manuscript.

### Grant Disclosures

The following grant information was disclosed by the authors:
Beijing Hospitals Authority's Ascent Plan: DFL20240603.
National Natural Science Foundation of China Youth Program: 82100397.

### Competing Interests

The authors declare that they have no competing interests.

### Author Contributions

- Xinyu Zhang conceived and designed the experiments, performed the experiments, analyzed the data, prepared figures and/or tables, and approved the final draft.
- Jie Yang conceived and designed the experiments, authored or reviewed drafts of the article, and approved the final draft.
- Yilin Li conceived and designed the experiments, authored or reviewed drafts of the article, and approved the final draft.
- Yulin Li conceived and designed the experiments, performed the experiments, analyzed the data, prepared figures and/or tables, authored or reviewed drafts of the article, and approved the final draft.
- Guoqi Li conceived and designed the experiments, authored or reviewed drafts of the article, and approved the final draft.

### Data Availability

This is a literature review.

## Supplemental Information

Supplemental information for this article can be found online at http://dx.doi.org/10.7717/peerj.19532#supplemental-information.

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
