# Peer review of "Role of hypothalamus function in metabolic diseases and its potential mechanisms"

_PeerJ, doi:10.7717/peerj.19532_

## Round 0.1 · original submission · Major Revisions

While both reviewers recognize the merit in this review and are keen to support its publication, both have clearly raised substantive issues that require your careful and detailed attention.

Their suggestions are elaborated in the reviews below, and I ask that you pay careful attention to all points raised and cover all new material suggested.

Note that I regard these changes are both essential and extensive; this should not be a 'quick-fix'.

Rather, it requires cogent and careful reevaluation of the topics covered, addressing the areas deemed to be lacking in depth, and in some key aeras new topics to be introduced. You should show clearly your PRISMA diagram (as a supplemental figure if you wish) explaining how you sourced and scanned the literature and identified the key studies you have discussed. This is important as both reviewer-1 and I agree that some key areas need to be covered which are not.

Reviewer 1 ·

Basic reporting

In this review, Zhang and colleagues aim to review neuroendocrine roles of the hypothalamus, and its implications to metabolic disorders. Although there are ample literature on the topic, it is an ever evolving field with substantial societal interest, given that obesity and T2DM is a comorbidity to many debilitating disorders.

The structure of the paper is adequate. That said, the authors address their targeted readership to this review, which could be reconsidered, given the scope of the journal includes broad and cross disciplinary research. If the decision to keep these statements is to be made, I would heavily advise on restructuring the introduction in a short paragraph and a section of "scope of review".

The language is fitting, but there are some points that could be revised throughout the manuscript, especially regarding the flow of the manuscript. For example lines 59-60: "The hypothalamus, through its neuronal populations, including the arcuate nucleus (ARC)60 and the paraventricular nucleus (PVN), senses and responds to the body’s energy status, affecting appetite,61 energy consumption, and overall metabolism through a series of complex neural circuits". Which could be read as: Hypothalamic neuronal populations, including ARC-residing and PVN-residing neurons, sense and respond to.. . Another example is the the Arc section, which starts with the non human primate study, but could nicely be finished with it instead, providing a valuable link to the PVN.

I believe the biggest flaw of the review is the lack of covering translational and human research, especially because of the statement that the review aims to inform to the medical community as well. In section 2, I provide some references that highlight the role of hypothalamic malfunctioning in humans, contemplating neuronal and glial features. Finally, I strongly recommend incorporating genetic forms of obesity in these (e.g. variants in MC4R, Prader-Willi Syndrome).

Figures are comprehensive and clear. However, the authors place the thalamic PVN instead of the hypothalamic in the design; this should be changed. In fact, oxytocin as an orexigenic mediator is inaccurate. Since the section on the PVN is mainly focused on MC4R, this should also be consistently present in the figure.

Experimental design

The description of the methodology and the execution are sound, and performed in a . As previously mentioned, I believe that the proposed scope of the review asks for the incorporation of seminal works on the hypothalamic dysfunction, and as well human-centered manuscripts as well. Below, the PIMID of some of these productions that the authors should evaluate to incorporate.

The review would benefit with the inclusion of genetic forms of obesity, with some supporting references also mentioned.

Seminal works in hypothalamic dysfunction in experimental animals PIMIDs: 15044184, 16002529, 11448938, 25233153, 34823066, 22201683,

Human hypothalamic dysfunction PIMIDs: 37467315, 31002795, 27568547, 33137293, 22438814, 30201275, 29736023, 27468060, 36170368, 32780722, 32814716, 9620771, 34848489, 24024123, 37400893, 26672638, 10486419, 27468060, 22492775, 27393312.

Genetic forms of obesity PIMIDs: 38556574, 29590610, 32692746, 9771698, 9771699, 9620771, 26179253, 19696756, 30677029, 32492392, 23869016, 18454146.

The review is logically organized. I would recommend an expansion of the PVN section, especially because there is ample literature. But I understand that the vastness of ARC-related literature should also be covered. The authors should consider to include the neuropeptidergic identity of PVN-residing neurons (e.g. Oxt) in its section.

Validity of the findings

The topic is of broad interest, and the review does cover important aspects of the hypothalamic "topography" dysfunction in metabolic disorders.

It is important to mention that the inclusion of the human centered research, and genetic components of obesity, would bring novelty to the manuscript, providing interesting perspectives for the field. This would differentiate the review from many others in the topic.

It is noteworthy to mention to the authors that the compilation of papers mentioned in previous sections are not mandatory, and the authors have agency to debate which literature can enrich their work.

Additional comments

Line 150: this cell metabolism energy described was specific in the diabetic condition.

The authors are invited to review their statements throughout the manuscript, to ensure no overstatements or simplifications of the actual results are made.

Reviewer 2 ·

Basic reporting

Overall, this is an interesting piece of work, but some strange phrases should be addressed.

1. line 38-39: better understanding of the central role of the hypothalamus in the mechanism of regulatory dysfunction. "regulatory dysfunction"
2. line 51 - the increase in obesity, diabetes, - add prevalence
3. line 69: the latest advancements in hypothalamic function - this should be phrased as our understanding of hypothalamic function (the function hasn't changed, but our knowledge has).
5. The introduction is quite wordy and should really cover what the authors are addressing in the review, not repeating who this article will be of benefit to.
6. Line 139 - these nuclei "house" neurons? I think the stated neurons are located within these regions, but to "house" them is a strange terminology.
7. Line 160- NPY plays an essential role in the short-term acute effect of AgRP neurons (should be activation of AgRP neurons).
8. Line 182 - POMC neurons exhibit complexity in controlling glucose homeostasis - this should be expanded upon.
9. Line 213 - alsoplay should be also play
10. Line 236-7 "Furthermore, insights into the modus operandi of VMH neurons in regulating food consumption have been unveiled", rephrase this for a scientific journal
11. Line 321-clearing ? PDK2 in astrocytes, I have never heard of clearing a gene. Remove this word
12. Line 358 - The Reverse Influence of Peripheral Organs on the Hypothalamus: I am not sure this heading fits the paragraph - remove the phrase reverse.
13. LInes 374 - to 388 The role of fat or adipokines is a very important area of research, and although touched upon, it is lost in this large paragraph. I would suggest a subheading to emphasise its importance. Add the section on leptin to this. The leptin story has been around for many years, and to date, targeting this pathway has not proved beneficial as individuals who are obese become leptin resistant, so I am not sure this pathway is an "inspirational" target.
14. Line 400-401 - Sh2b1 for gene, SH2B1 for protein.
15. Line 431 - remove which - GLP-1 receptor agonists emulate the physiological actions of GLP1.

Experimental design

This is a systematic review but does not state how many articles were reviewed or finally accepted for final analysis. The review does not seem to follow PRISMA guidelines (Preferred Reporting Items for Systematic Reviews and Meta-Analyses). Is there a reason for this?

Validity of the findings

1. Some areas need to be expanded upon - Line 164-5 AgRP neuron activity also affects insulin sensitivity and glucose metabolism. This is a huge area of research that is limited to a single line.
2. If talking about the TET23 gene, it would help to explain what this does and why loss of this gene might influence metabolic parameters - there is no mention that the TET family of proteins are involved in DNA methylation.
3. Line 203-4 - glucose sensing cells can perceive changes in blood glucose levels. This needs to be expanded upon - no mention of GE and GI neurons; what happens to these neurons in obesity/diabetes? There is a lot known about these neurons, but this is not mentioned.- this is relevant to the Ano4 expressing neurons that are GI neurons.
4. Line 214 - through the connections with other brain regions and peripheral tissues, this is too vague for this kind of article. What other brain regions and tissues? This detail is important and would help the authors describe the role of the PVN and different neuronal cell types.
5. Lines 232-236. The discussion of O-GlcNAc should be rephrased as there are multiple other functions of O-GLcNAC that haven't been mentioned that could have implications if this is to be targeted, as the authors proposed.
6. Line 313 - HFD promotes UCP2 expression, causing microglia activation and pro-inflammatory factor release - which cytokines are released? this sentence needs clarification.

Additional comments

Overall, this is an interesting review that brings together several aspects of the literature to try and better understand the role of the hypothalamus in energy homeostasis. However, it lacks some key research techniques that have helped our understanding of the hypothalamus, including PET scanning to identify regions of the brain activated (or inactivated) in those with metabolic disorders, what happens to hypothalamic signalling following bariatric surgery and also following extreme dietary restrictions. All of these interventions may lead to changes in nutrient sensing that is not mentioned. Furthermore, the authors don't mention GE and GI neurons that play an important role in diabetes and, in particular, the counter-regulatory response to hypoglycaemia, seen in those with type 1 and insulin-dependent type 2 diabetes.

---

## Round 0.2 · Major Revisions

As you will see, both reviewer-1 (and I) still have concerns about your selection of the literature and the lack of transparency on the criteria you used.

Previously, I noted: You should show clearly your PRISMA diagram (as a supplemental figure if you wish) explaining how you sourced and scanned the literature and identified the key studies you have discussed.
Although there is a PRISMA diagram, it raises questions - specifically, how automation tool removed in appropriate references (and on what basis), what exclusion and inclusion criteria did you use for this? - your description in the text is not informative. There needs to be a far more extensive and clear approach explaining how you chose these studies.

Reviewer-1 has further suggestions.

I also was clear that a substantive and careful re-write was required. I think that this paper would also benefit from careful English proof-reading. There are places where the text is rather convoluted.

**Language Note:** The Academic Editor has identified that the English language must be improved. PeerJ can provide language editing services - please contact us at [email protected] for pricing (be sure to provide your manuscript number and title). Alternatively, you should make your own arrangements to improve the language quality and provide details in your response letter. – PeerJ Staff

Reviewer 1 ·

Basic reporting

In general, the authors have integrated the feedback provided into their manuscript. The incorporation of human-centered literature brings novelty to the review, and is a differential among already extensive existing literature.

The structure provided is logical. However, the manuscript would benefit from language revisions, specially to betterment of the flow. In some cases, the authors also make the reading difficult by the inclusion of molecular mediators of hypothalamic neuronal function, which are not necessary to fully understand the phenotype/physiological process (eg protein tyrosine phosphatase δ receptor in AgRP neurons). These inclusions made the text tardy, and can frustrate the readership.

I would strongly recommend a language revision, which can further improve the quality of the text.

Figure 2: the old figure is provided in the concatenated file, whereas the revised figure is present in the rebuttal.

Experimental design

Questions about the methodology were raised on the first round of revision, and the authors provided further details; and presumably executed a new search.

I am not confident that this search yield the most convincing results. It is reasonable top speculate that the numbers of manuscripts that were to be screened would be much higher (e.g. animal models of PWS would naturally appear in the search - and could be even excluded of the analysis - but nothing is said. Important works on GLP1 agonists also do not appear in the screening although expected - eg PMID: 25089000, 21693680).

The authors are invited to revise this aspect of their manuscript, especially because their results do not seem to reflect the existing literature.

Validity of the findings

Please see above; other than that no concerns

Reviewer 2 ·

Basic reporting

The authors have addressed most of the comments raised in my original review. There are a few typographical errors and some words that have been used incorrectly (emockate - should be replaced with mimic). The referencing needs to be updated as there is a combination of older references in curved brackets (x) with the new brackets [x].

Experimental design

The inclusion of the PRISMA diagram has improved the flow of the manuscript, as has the restructuring of the paper.

Validity of the findings

The conclusions are valid within the context of the manuscript

Additional comments

The manuscript has been greatly improved; however, there remain several typographical errors that need to be addressed and referencing must be checked as it appears that there are two formats in the same manuscript.

---

## Round 0.3 · accepted · Accept

Thank you for clarifying the remaining issues and attending to the outstanding issues. I am delighted to recommend acceptance now.